# Does the Human Right to Healthcare Apply Universally? A Contribution from a Trauma Therapeutic Perspective

**DOI:** 10.3390/ijerph20156492

**Published:** 2023-08-01

**Authors:** Bernd Hanewald, Daniel Berthold, Markus Stingl

**Affiliations:** 1Center for Psychiatry and Psychotherapy, University of Giessen, 35392 Giessen, Germany; 2Department for Medical Oncology and Palliative Care, University Hospital of Giessen and Marburg, 35392 Giessen, Germany

**Keywords:** post-traumatic stress disorder, trauma sequelae, flight, structural violence, human rights

## Abstract

Access to the best possible healthcare is a fundamental human right. However, the provision of medical treatment is not only dependent on the actual treatment options available and the type of illness to be treated but is significantly influenced and restricted by structural and legal conditions. This is particularly evident in the case of refugees and other groups such as the so-called “paperless”, whose access to medical treatment is de facto seriously impeded or denied altogether. At the same time, these individuals are particularly vulnerable to the development of mental illness for a variety of reasons. Refugees in particular often suffer from trauma sequelae, resulting in a broad range of impairments. Based on a case study of a refugee woman living in her host country, the interactions between mental illness and limited psychiatric/psychotherapeutic treatment options due to legal restrictions are analyzed from a medical perspective. Her initially only medically oriented treatment was insufficient to mitigate the consequences of these restrictions. As it was a protracted treatment process, the legal aspects of her case therefore also had to be decisively considered. This case study shows that the human right to the best possible healthcare can be considerably restricted by structural requirements, which, in the case of sequential traumatization and severe illnesses with suicidal tendencies, can be labelled as structural violence.

Among refugees, there are complex interactions between mental health, actual treatment options and legal frameworks. The objective of this essay is to discuss these interactions from a human rights perspective, exemplified by a healthcare provision situation in a host country.

## 1. Prevalence of Mental Illness among Refugees

The above-mentioned limited medical treatment options for refugees contrast with the high prevalence of mental disorders. Refugees and asylum seekers are highly vulnerable groups with an increased risk of developing mental diseases. According to international studies, 30% to 70% of refugees suffer from trauma sequelae disorders [1,2,3,4,5]. In a study sample of 125 subjects from a local initial reception center for refugees and 116 refugees living in community shelters in their host country, the authors of a screening study [6] found clear evidence of the presence of trauma sequelae disorders in 65.9% and 80.0% of these populations, respectively. Unaccompanied minor refugees, who were examined during the initial medical examination, also showed a high prevalence of trauma sequelae disorders depending on their regions of origin (the positive screening values of the origin-related samples were between 42.9% and 62.3% [7]).

These trauma-related psychological sequelae manifest not only as post-traumatic stress disorders (PTSD) but also as other severe mental illnesses, such as depressive disorders, anxiety disorders, addictive disorders, personality disorders or somatoform disorders [8]. The underlying traumatic experiences of refugees usually originate in their home country, e.g., from war-associated trauma or persecution, but may also be exacerbated during their flight by experiences of extreme physical or sexual violence, hunger, the loss of relatives or friends, arrest or torture. In addition, subsequent experiences in their destination country can perpetuate or exacerbate refugees’ existing psychological complaints and symptoms, which can profoundly affect their expectations and hopes for safety and more stable living conditions, which were overriding motives for leaving their home country.

## 2. Framework Conditions for Medical Treatment of Refugees

The United Nations Universal Declaration of Human Rights of 1948 emphasizes access to the best possible healthcare as a fundamental human right. In the same year, the World Health Organization (WHO) was founded, whose constitution states that “The enjoyment of the highest attainable standard of health is one of the fundamental rights of every human being without distinction of race, religion, political belief, economic or social condition” (https://www.who.int/news-room/commentaries/detail/health-is-a-fundamental-human-right. Accessed on 17 March 2023). The Geneva Declaration, adopted at the second General Assembly of the World Medical Association, emphasizes, as a precondition of medical ethics: “The health of my patient will be my first consideration. […] I will not permit considerations of age disease or disability, creed, ethnic origin, gender, nationality, political affiliation, race, sexual orientation or social standing or any other factor to intervene between my duty and my patient.” (It should be noted here that there are no human races, as genetic variations within the human population are not clearly defined, and there is no biological basis for classifying humans into different races. The term ‘race’ in reference to humans is therefore meaningless, but rather a social myth, as there is no objective classification based on biological characteristics that would allow for a sharp separation of groups. Instead, human diversity is better described using the terms ‘ethnic groups’ or ‘population groups,’ which can reflect cultural and geographic differences (UNESCO 1950, The Race Question. https://unesdoc.unesco.org/ark:/48223/pf0000128291. Accessed on 17 March 2023)).

More than seventy years after the Universal Declaration of Human Rights and the Geneva Declaration, as well as the founding of the WHO, it may seem surprising that the human right to healthcare requires special emphasis. This is particularly obvious in the context of the treatment of refugees, and these declarations are now even cited to legitimize medical action in particular. Indeed, the insecure residence status of refugees in particular may impair refugees’ access to adequate healthcare [9]. The medical treatment of refugees in their host country is initially subject to Section 4 Article 1 of the Asylum Seekers Benefits Act, which regulates the entitlement to benefits of asylum seekers and certain other groups of individuals in their host country. According to this, asylum seekers are generally entitled to receive benefits to secure their livelihood. These benefits typically include both in-kind and cash benefits. The amount of benefits is based on the necessary requirements to ensure a level of subsistence.

This formally stipulates that only acute illnesses and pain conditions may be treated. In addition, according to the opening clause of Section 6 of the Asylum Seekers Benefits Act, “other benefits may be granted in particular if they are indispensable in individual cases to secure subsistence or health, necessary to meet specific needs of children, or required to fulfill an administrative obligation to cooperate. Benefits are to be provided as in-kind assistance, and in exceptional circumstances, as monetary assistance.” One of the serious consequences of this legislation is that, under a narrow interpretation, chronic illnesses must initially remain untreated or may only be treated when acute decompensations have occurred that are, in principle, avoidable. From the point of view of medical ethics, it is difficult to understand that the decision as to whether an illness exists according to Section 4 Art. 1 of the Asylum Seekers’ Benefits Act and whether the costs of medical treatment are to be covered must be approved by authorities before patients are put into contact with doctors. This decision is therefore not made by medical staff, but by employees of authorities without medical training and thus by medical laypersons. In addition to the aforementioned, the acute aggravation of diseases through non-treatment that are in principle avoidable, a further consequence of this regulation is that after an official refusal of the necessary treatment, patients are forced to visit the emergency outpatient departments of clinics or to use other emergency medical services. Thus, from the point of view of those affected, the presentation of acute symptoms in hospital emergency departments or during outpatient medical emergency services is often the only way to obtain necessary medical treatment. Nevertheless, in these cases, one’s history of treatment in the standard care system would often be indicated, provided that the corresponding access possibilities exist. Frequently, this circumstance causes mutual frustration and tension among the patients concerned and the staff in already overburdened emergency departments. Against this background, whether the necessary treatment takes place in an emergency department—even if there is no emergency in a narrow sense—is ultimately dependent on several imponderables: How convincingly can the concerned refugees present their acute need for treatment, given the language barrier that often exists? How strictly are the framework conditions of the Asylum Seekers’ Benefits Act interpreted by potential treatment providers and institutions? To what extent does the workload of the respective outpatient clinics allow for discretionary leeway?

In summary, it must be stated that in practice, the medical treatment of refugees is not primarily based on their medical history and needs, but is dependent on their residence status, further legal implications, administrative assessments of their disease and their health status, as well as other incalculable framework conditions, even though the indication of medical treatment should always be based on individual decision-making processes, for which trust is a basic prerequisite, and which must not be undermined by administrative obstacles.

## 3. Post-Traumatic Stress Disorder Is a Frequent Trauma Sequelae Disorder

PTSD is a common disorder following traumatizing life events. It represents a delayed or protracted response to a catastrophic stressful event or a short- or long-term extraordinarily threatening situation that would cause deep despair in almost anyone [10]. In a traumatic situation, a person is flooded with aversive stimuli and is no longer able to extricate themselves from the situation without others’ help. They are no longer able to use innate archaic behaviors that ensure survival, such as “fight” or “flight” responses, and experience themselves as helpless, powerless and at the mercy of others. This extremely stressful situation is resolved mentally in the short term by so-called “peritraumatic dissociation”: the traumatized is incapable of action and “freezes”, similar to the stasis reflex in certain animal species. In this process, the actual experience is fragmented, i.e., the associated emotions and bodily feelings are split off from the experience. The problem with this psychological protective mechanism is that the traumatic experience cannot be processed as a complete story with a beginning, middle and end and thus cannot be transferred into the personal narrative as a complete whole. Parallel to memory gaps, “memory islands” can also develop; after a traumatic situation, individual details are often still remembered very precisely, while large parts of the experience remain completely inaccessible to one’s memory. These psychological consequences of extreme threats can be explained neurophysiologically. Three brain structures are essentially involved: the thalamus, the amygdala and the hippocampi. During “normal” information processing, the thalamus, as the stimulus reception and processing station, relays new information to the amygdala, the hippocampus and the prefrontal cortex. This processing includes, among other things, the classification of incoming information by its context and a conscious evaluation of this information before subsequent action planning. The hippocampus, as part of one’s conscious, explicit memory, represents a kind of “memory store” and is essentially involved in the fact that events from the past can be biographically remembered and verbalized. The amygdala, on the other hand, plays a central role in the one’s unconscious emotional response to potential danger. As experiences of danger are associated with strong aversive emotional arousal, there is a corresponding increased activation of the amygdala in the immediate processing of the experience, in simplified terms. This ultimately causes the increased release of the stress hormones adrenaline, noradrenaline and cortisone from the adrenal cortex. The increased release of these stress hormones in turn leads to the hippocampal system no longer being able to store or integrate everything experienced into one’s personal biography. Often, only fragments of what has been experienced are stored; the above-mentioned partial amnesia as well as “memory islands” result. Furthermore, the increased activity of the amygdala during the traumatic event can trigger the formation of so-called “implicit memories”. This means that although one can still remember that something happened, they can no longer fully remember what happened. Memories can then consist, for example, only of the associated physical reactions (“body memories”) without a context for the content (the “what”) (e.g., pain as a body memory of a traumatic situation without a conscious link to the triggering event). Implicit memories are often activated by triggers (see below) and show up in the pain experience similar to the pain during the traumatic event itself or in other bodily vegetative reactions, such as heart palpitations, restlessness, sweating or trembling.

Often, with a time lag from the traumatic experience, there is an involuntary reliving of the trauma in thoughts, intrusions, nightmares or flashbacks, which is triggered by a cue that has a similarity to the original traumatic situation. There is often a feeling of numbness, emotional dullness, indifference to others, apathy and anhedonia, i.e., a comprehensive loss of joy in life. In addition, anxiety symptoms and symptoms of depression with suicidal ideation, hyperarousal, an increase in vigilance and being easily startled may occur. Upon recollection of the traumatic event, emotional and physical symptoms are immediately triggered. Therefore, sufferers avoid activities and situations that could be effective as triggers and evoke memories of the trauma. Affected individuals often suffer from sleep disturbances and may also experience irritability and difficulty concentrating. The aforementioned symptoms follow the traumatic experience with a latency that may last weeks to months, but rarely exceeds six months [10].

The COVID-19 crisis undoubtedly worsened the already precarious situation of refugees. The closure of borders resulted in many refugees being trapped in transit or reception facilities, with their asylum claims unable to progress, which added additional uncertainty and delays for people who were already suffering from traumatic experiences and had to leave their homes out of necessity. Furthermore, in some countries, assistance and support measures for refugees have been reduced or restricted. Scarce resources and the prioritization of the national population in the midst of the pandemic led to refugees often having less access to healthcare, education and other essential services [11,12]. Moreover, economic constraints and business closures during the pandemic led to layoffs and income losses for many people, including refugees. Many refugees work precarious jobs which were particularly affected by the crisis and were laid off due to business restrictions. This has exacerbated their existing economic insecurity and makes it more challenging for them to secure a decent livelihood. The deteriorating conditions at borders during the pandemic also led to an increase in irregular migration and a heightened use of smuggling networks. With the closure of legal migration routes and the lack of safe alternatives, many people had no choice but to resort to dangerous and unsafe routes in search of protection [11]. To show how the structural and legal framework conditions in one’s host country can influence the course of their treatment for trauma sequelae, a typical case vignette of a trauma patient who has fled her country is illustrated.

## 4. Case Study

Ms. B., a 33-year-old female patient, came severely depressed with latent suicidal thoughts to a university hospital in her host country for the first time in June 2012. On admission, she suffered from a major depressive episode with psychotic symptoms, a somatoform pain disorder, post-traumatic stress disorder and a dissociative disorder. Mrs. B. spoke only a few words of the host language, but when in contact with staff who spoke her native language, she was able to engage in treatment very slowly. Her depressive symptomatology included a depressed mood, circling thoughts, inner restlessness, lack of interest, social withdrawal, suicidal thoughts, lack of appetite, sleep disturbances and pronounced joylessness as well as strong fears, panic attacks, nightmares, flashbacks, physical pain and tension.

Regarding her biographical background, she reported in summary that she had fled to her host country with her husband and a son (who was one year old). The journey from her country of origin to her host country took only a short amount of time, and there were no incidents of imprisonment, human trafficking or other significant hardships during their escape. In her host country, they had applied for asylum; their first application had been rejected. Until they were admitted to the clinic, the family had lived in the host country without a secure residence title. The couple had three more children, who were born in the host country.

Over the years, Mrs. B. developed increasing fears of possible deportation back to the place of origin for her traumatic experiences, as well as increasing psychological problems. She reported sleep disturbances, physical pain with no apparent organic cause, anxiety and jumpiness. Toward her husband and children, she often reacted aggressively and unfairly, which in turn increased her feelings of guilt. Intrusions and flashbacks from her time in her home country imposed themselves. Yet she had so far refused psychiatric help for fear of being thought crazy and stigmatized. The patient sought medical care at the clinic in her host country nine years after arriving in the country. It was only when she began experiencing suicidal thoughts that she sought help, with the assistance of a Kurdish medical colleague specializing in neurology, who referred her to the clinic.

The patient reported that she grew up with eight siblings on a farm in a poor rural region. Members of her family had been fighters in a resistance movement. There had been repeated reprisals and physical and verbal violence by the police against the family. One day, soldiers brought the body of her brother, with whom she had had no contact for several years. She had just been outside with her family on a summer day when military vehicles drove up, including a pickup truck with four soldiers. The soldiers threw her brother’s body face down from the back of the pickup to the ground and shouted, “There’s your dog!” Her father ran to the body, threw himself on top of her brother, turned him over, yelled out and called her brother’s name. The soldiers kicked both her father and her brother’s corpse, and her father suffered several broken ribs. When they realized that their brother had died, they were “out of their minds”. The memory of the moment when her father turned over the corpse weighs on her the most. When she thinks about it today, she still feels a massive amount of fear, trembles and feels powerless, “as if the brain stops”. She is very afraid that something similar could happen to her children in their country of origin.

In interpreter-assisted therapy sessions over a period of months, she reported further threatening and almost life-threatening events with physical and sexual violence which had recurred since her childhood, to which she had been exposed both as a victim and as a witness, and because of which her brother had joined the resistance as a fighter due to the “family disgrace”. During the interviews, she repeatedly stated that she had been degraded and that it was “better to have been beaten to death than to have experienced such things.”

During her treatment, further flashbacks, intrusions and nightmares occurred at times. Ms. B. suffered from visual hallucinations, so she “saw” her brother, who had been killed, covered in blood in the ward and also repeatedly saw police officers in the clinic, whom she was afraid of and hid from. Although she withdrew a lot, she managed to maintain contact with the treatment team during these phases. We evaluated the visual hallucinations that occurred during the flashbacks as psychotic experiences in the context of post-traumatic stress disorder and treated them with medication. Psychotic experiences can occur in post-traumatic stress disorder, so the psychotic subtype of post-traumatic stress disorder was discussed [13,14,15]. It must be noted at this point, however, that the previous classification systems have reached their limits in the case of these complex disturbance patterns. In the case of extreme traumatization, there is a risk of overlooking the psychotic qualities of experience, just as, conversely, traumatic previous experiences can be overlooked in the case of psychotic experiences existing in the foreground [16].

At the beginning of the treatment, Ms. B. showed pronounced fearful mistrust towards her treatment team and strong insecurity. Fear and mistrust made it difficult to establish and deepen the therapeutic relationship; however, they are understandable against the background of her traumatic previous experiences, uncertain residence status for many years and the repetitive nature of having to request leave to remain, i.e., she could stay for only a limited time and had to repeatedly apply for extensions. The latter complicates the treatment of post-traumatic stress disorder and is understood by Henningsen [17] as a “doubling of the paranoid attitude towards the situation.” Since her fear of deportation made treatment almost impossible, the family’s lawyer and the refugee law clinic at the local university’s department of law served in an advisory capacity. The statements of the clinic were criticized by the responsible asylum authority in terms of their content, and the symptoms shown by Mrs. B. were doubted. In addition, reference was made to treatment options for the disease in her country of origin, but in our view, this did not take into account the risk of retraumatization. In parallel, Ms. B.’s condition deteriorated again during treatment. Letters from the authorities and an announced visit by the Medical Service of the Health Insurance Funds to check the necessity of her inpatient treatment led to a crisis-like escalation of her symptoms. Since targeted and, in the present case, urgently indicated trauma therapy requires not only reliable and sustainable therapeutic relationships but also external security, the patient could only be treated supportively against the background of imminent deportation. The situation was difficult to bear not only for Ms. B. but also for the treating team, who experienced increased helplessness and powerlessness. Her lack of residence status almost completely eclipsed all other aspects, e.g., medical, social, cultural or religious aspects. The situation was further complicated by requests from her health insurance company to end the inpatient treatment promptly. This only changed when the case was clarified by the courts. Thus, in September 2014, the Administrative Court ruled that Ms. B. should be granted a ban on deportation under Section 60 [7] sentence 1 of the Residence Act (“considerable concrete danger to life, limb or freedom”). This was the basis for the patient now being treated with specific trauma therapy interventions and, after a certain period of therapeutic work, being discharged and transferred to outpatient treatment. However, her protracted stay in the clinic had considerable psychosocial costs, including increased distance from her husband and children with a consequent deterioration in her children’s school performance. In an impressive letter, even the class teachers of her school-age children turned to the responsible authorities, since the negative effects of the mother’s illness on the children’s school performance and well-being could no longer be overlooked. Furthermore, the husband had to temporarily stop his professional activities to adequately care for the children and to compensate for the mother’s absence at least to some extent.

Ms. B. was in inpatient psychiatric treatment for 25 months, with only one brief interruption. The attempt to release Ms. B. from the inpatient setting and to treat her as an outpatient prior to the court’s determination of the deportation ban failed due to her increasing fear of deportation with renewed suicidal tendencies, and Ms. B. had to be readmitted to the clinic.

## 5. Features in the Treatment of Refugees

Trauma sequelae in refugees can result from direct war experiences to which the affected persons were exposed as civilians or in which they were involved as military personnel. The latter requires an additional analysis of one’s own victim and perpetrator roles in therapy. Experiences of violence in the course of civil-war-like conditions are just as frequently causes of trauma as the experiences of torture, imprisonment or death threats in one’s home country. On escape routes, victims are then exposed to further dangers, such as hunger, imprisonment and physical and/or sexual violence, so the risk of further traumatization increases. According to a 2015 position paper by the body of psychotherapists in a host nation (https://api.bptk.de/uploads/20150916_bptk_standpunkt_psychische_erkrankungen_fluechtlinge_a9eecbf8c9.pdf. Accessed on 17 July 2023), 70% of refugees have witnessed violence against third parties, half have seen dead bodies and 55% have been victims of violence themselves, with 43% having been tortured. According to a study by Jakobsen et al. [18], the asylum seekers interviewed had experienced an average of nine traumatic events. Trafficking organizations blackmail family members remaining in the home country with further demands for money by repeatedly arresting those on the run and demanding ransom payments. This extends individual traumatization to entire family systems. The steady succession of stresses is also referred to as cumulative traumatization [19] or sequential traumatization [20], which is associated with symptom amplification and worsens one’s prognosis for treatment. As evident in the case study, existential fears of deportation interfere with psychotherapeutic and psychiatric treatment or make it nearly impossible. Conventional therapeutic approaches that ignore refugees’ legal situation in their treatment may overlook their fundamental needs; thus, an appropriate expansion is necessary here. This is evident when the initial stabilization of a patient is achieved during treatment, but continued and supportive internal security cannot be established due to the lack of external security provided by their insecure residency status. This makes further trauma therapy considerably more difficult. Contrary to the evidence of the effectiveness of some psychotherapeutic methods in the treatment of refugees, from our practical experience, we are critical of the medium- to long-term success of trauma therapy treatments that include confrontational elements if refugees’ safety cannot be granted as a basic prerequisite. The sometimes-long period of uncertain residence status as well as the uncertain asylum procedure are undoubtedly factors that can perpetuate refugees’ mental illness, significantly impair their psychotherapeutic options and reinforce previous traumatic experiences [7]. Against this background, the diagnosis of post-traumatic stress disorder also does not seem appropriate, as it is not really a “post”-traumatic situation [21], but rather disease-promoting and disease-maintaining factors persist.

In addition to the influencing factors outlined in terms of asylum law [9,22], several other conditions cumulatively complicate the treatment of refugees and asylum seekers: difficult housing conditions, loneliness, helplessness, the lack of fulfilling and meaningful daily activities, the loss of social structures, grief, the loss of family and guilt. Often, their flight was financed by family members with a poor initial economic situation; however, this is often linked to an explicit or implicit expectation that the refugee should financially support those left behind or seek family reunification in the future. If these expectations cannot be fulfilled, e.g., about children or spouses who have remained in precarious living situations, the refugee may feel a great amount of guilt. These factors are to be understood as acutely effective stressors and make psychotherapeutic treatment even more difficult; they can worsen existing psychological problems or promote the development of additional psychological disorders. On a structural level, language difficulties and the often-necessary use of interpreters further complicate their treatment, as do uncertainties on the part of the persons concerned about the procedures of the asylum process. Only rarely do the persons concerned have the opportunity to seek legal advice before their first hearing at the BAMF. As described above, access to medical care is difficult or impossible. The situation of those affected in initial reception facilities is often characterized by a lack of opportunities to withdraw, a lack of daily structure, poor living conditions and the need to adapt to, for example, different eating habits and cultural conditions in the destination country.

The interactions between refugees and host communities can be reflected in various ways. On the one hand, migrants have non-economic potential which influences their interactions with their host society. This can include cultural diversity, language skills, knowledge and new perspectives [23]. This diversity can contribute to an enrichment of the host society and foster the exchange of ideas and experiences. On the other hand, it can also lead to misunderstandings or conflicts when different cultural norms, values or communication styles collide. The arrival of forced migrants means that individuals and groups are thrown together in the host society and must learn to interact in new ways [24]. This can lead to tensions, particularly when initially unconditional ethical hospitality turns into conditional hospitality. The reception and integration of refugees can present ethical-humanitarian, socio-economic and political demands that are not always easy to reconcile. The concept of hospitality often entails an ambivalent perception full of contradictions. Hosts may feel threatened by the presence of refugees for various reasons, such as competition for limited resources, such as healthcare services [24]. Additionally, tensions can arise when refugees are unwilling or unable to leave, even if the original reasons for their visit are no longer valid. It is important to recognize that the relationship between a host and guest is not necessarily stable over time and does not exist in a binary state. The dynamics and interactions between refugees and host communities can change over time and are influenced by various factors, including political developments, socio-economic conditions and individual experiences [24]. The reflections above illustrate that the tensions between refugees and host communities are multifactorial and depend on a variety of factors. A nuanced approach is necessary to adequately understand and address the challenges and potential of this relationship.

For those providing treatment, the influences of social, political, legal and economic conditions on the mental health of refugees and their interaction with treatment options and treatment outcomes are an everyday problem. Looking at the problem of language barriers, for example, it is clear that providers often have little experience in dealing with interpreters and therapy. In this context, it is necessary that the involvement of interpreters is supported in principle by the treatment team. Finally, the financing of the interpreter service must be guaranteed, which is not always clarified in standard care, otherwise these bureaucratic hurdles must be overcome. This puts an extreme amount of strain on the framework of outpatient psychotherapeutic care and makes many practitioners shy away. Cultural aspects in the treatment of refugees must also be considered as a possible source of problems; it should not be overlooked what kind of treatment and assistance a patient needs most urgently in individual terms. Too much emphasis on or consideration of the presumed “culture” of the patient as a background to an understanding of the emotional content of their experience and their exhibited behavior runs the risk of producing one-sided stereotypes and can significantly jeopardize an empathic, individualized approach to treatment [25,26]. Moreover, diagnostically challenging is the fact that although the affected individuals often suffer from mental health problems or psychiatric illnesses, other symptoms often lead to acute hospital admission. For example, refugees often have somatic complaints such as headaches, chest pain, stomach pain or sleep disturbances and report any psychological complaints late or not at all [27], which makes an adequate diagnostic assessment difficult. Fear of stigmatization due to mental illness plays a significant role here [14].

Due to the aforementioned special features and resulting uncertainties, the treatment of refugees in practice usually remains rather superficial and is limited to short-term stabilization within the limits presented. Thus, the treatments offered are often ineffective in the medium to long term, associated with negative disease courses and increased treatment costs. Consequently, against the background of these comprehensive peculiarities and interactions, it is necessary to counter them with specific treatment concepts to be able to guarantee the best possible healthcare. In practice, the central requirement in the treatment of refugees is to look for possibilities that open up efficient, individualized and patient-oriented therapeutic interventions, which provide the treatment team with security and orientation in challenging situations and which contribute to increased treatment satisfaction of both the patients and treatment providers [7].

## 6. The Universality of Human Rights—Relevance for Practice

The right to health is a fundamental part of human rights and of a life with dignity (http://www.ohchr.org/Documents/Publications/Factsheet31.pdf. Accessed on 17 March 2023). This results in the right of entitlement to access to good, comprehensive healthcare in the event of illness, to achieve the “highest attainable standard of physical and mental health” for an individual. This implies that although diseases, impairments or disabilities are aspects of human life that are to be accepted, their prevention, reduction or cure are nevertheless the goals to honor the right of entitlement—this must be emphasized even more because, in the past, the idea of a “world without weakness and disease” was perverted, e.g., as in eugenic concepts [28]. The latter dramatically illustrates how the implementation of the right to health is dependent on contextual factors. In addition to political decisions, other social determinants of health, such as the distribution of money, power and other resources, are operative at global, national, as well as local levels (https://www.who.int/health-topics/social-determinants-of-health#tab=tab_1. Accessed on 17 March 2023).

The current constellation of laws, legal practices and procedures that systematically prevent refugees from having the human right to access the healthcare system and thus the “highest attainable level of physical and mental health” can be understood as “structural violence”, following Farmer et al. [29]. Due to the described access difficulties to the medical aid system, depending on legal aspects, political decisions and the goodwill of medical laypersons who are entrusted with decisions to approve costs, as well as context-dependent contingencies, non-discrimination and universality do not entirely transfer to lived practice.

In addition to non-discriminatory access to medical care, health-promoting living conditions must be ensured as further social determinants of health. These include not only safe access to clean drinking water, safe food and adequate nutrition, but also healthy living, working and environmental conditions, as well as health-related information, education and gender equality (http://www.ohchr.org/Documents/Publications/Factsheet31.pdf. Accessed on 17 March 2023). The exclusion of individual groups from health services and the partial closure of resulting gaps in care through voluntary institutions, for example, run counter to a human right to healthcare that is binding under international law and the social covenant ratified by the host country. The medical care to be initially approved and thus granted by the Asylum Seekers Benefits Act is significantly less than the benefits catalog of statutory health insurance funds, although the latter is also already subject to the standard of what is medically required and necessary. This unequal treatment of individual groups makes people ill; in particular, it can result in acute and chronic illnesses that are preventable in principle, which can have a consecutive effect as an obstacle to their integration and undermine other guarantees enshrined in human rights, such as participation in social, societal and political life. The fact that medical treatment is not provided based on medical indications but based on legal implications and depending on one’s insurance status is also to be challenged from the point of view of professional ethics. Treatments are always the result of individual decision-making processes for which trust and a relationship are the absolute basics, which must not be jeopardized by bureaucratic obstacles.

In the above-presented case study of Mrs. B., was her inpatient treatment for more than two years necessary or even urgently required from a human rights perspective? Her residency and asylum law or the derived official practice dramatically intensified the consequences of the trauma experienced in her country of origin, which in turn also had considerable consequences in her social environment. This corresponds in essence to “sequential traumatization” [20], equivalent to a continuous subjective experience of powerlessness, helplessness and fear, which is absolutely counterproductive from a trauma-therapeutic point of view and stands in the way of mentally processing what has been experienced: the victims internalize the experienced structural violence against the background of their subjective traumatic experiences, so that the trauma remains present in the here-and-now. In our case study, due to the threat of deportation, the patient lacked the necessary external security for a long time to be able to benefit adequately from the trauma therapy treatment. The course of treatment for her mental illness based on sequential traumatization was seriously complicated by this structural violence, which led to a considerable prolongation of the course of the illness and furthered her and her family’s suffering, which was potentially avoidable. The effects of her insecure residence status meant that the patient had to be denied the basic right to the best possible healthcare, since the treatment options would have been incomparably better with a secure residence status.

On the other hand, structural violence does not work through anonymous forces that unfold autonomously in a Kafkaesque sense. On the contrary, laws, regulations and structures are created and put into practice by individual actors. Existing legal provisions allow for discretion and room for maneuver, which can be used in the interest of the persons concerned and their mental health—or not. In this case study, we were able to partake in the often not-easy discussion of the treatment process, which seemed hopeless for a long time, guided by our orientation towards medical ethics and human rights. Article 25 of the Universal Declaration of Human Rights served as our underlying axiom. From our point of view, we did not want to focus exclusively on psychiatric/psychotherapeutic treatment options in the patient’s treatment, but rather include legal aspects as well. Together with the refugee law clinic and a lawyer, we were able to explain the patient’s mental illness to the administrative court in detail, clearly and comprehensibly, as well as to anticipate and stipulate the potential health consequences to the court in the event of a failure to provide further treatment. In this way, further consequences of this structural violence, which consequently restricted her access to the best possible healthcare as her fundamental human right, could be reduced or avoided, and the sustainable stabilization and outpatient treatability of the patient could be achieved. The extraordinarily long and thus expensive course of inpatient treatment for the funding agency was legitimized by the pursuit of the best possible healthcare required by human rights, which was not initially available in the case study and could only be achieved through an appropriate intervention.

Deportations can have a considerable influence on the course of pre-existing mental illnesses—the deportation itself, refugees’ changed living conditions and their possibly inadequate treatment options in the next country can have a negative impact on mental illnesses. Temporary suspensions of deportation for medical reasons or exceptional leave to remain are usually not a way out in the case of mental illnesses, since against the background of continuing uncertainty, as shown above, adequate treatment is not possible and even promotes secondary illnesses—the person concerned must remain ill in order not to be deported. Behind this is not necessarily a conscious and intentional deception. Rather, the development and maintenance of symptoms through subsequent treatment lead to a reduction in their overwhelming fear of deportation and, at the same time, an imminent danger of further chronification. As a result, those affected are, to a certain extent, under a psychological compulsion to maintain their illness.

In the case of actual deportation, in turn, there is usually a fear that psychopathological symptoms will reoccur and even lead to retraumatization, and any treatment successes that have been achieved to date can be massively jeopardized. If the person concerned has been exposed to trauma in their country of origin, as in the case example above, repatriation is almost certainly associated with a considerable deterioration in the state of health of the person concerned, irrespective of the real threat that still exists at present, which the psychiatric treatment provider is usually unable to assess adequately. This follows from the internal logic of trauma sequelae: in the case of existing post-traumatic stress disorder, events that are directly or indirectly related to the stressful events can function as trigger factors that evoke memories of real traumatizations that have taken place or massively intensify them, for example, in the form of intrusions and flashbacks. From a professional psychiatric point of view, therefore, people cannot be treated in environments that stimulate intrusions and do not allow for avoidance behaviors. Against this background, treatment options in refugees’ home country are often omitted, quite independently of objectively available therapy options. The prerequisite for the adequate treatment of traumatized refugees is therefore a living situation in which those affected can feel relatively safe from renewed traumatization, and renewed contact with perpetrators or threats of violence and injury are largely excluded.

Valid, criterion-based and professionally responsible diagnostics are required to recognize a refugee’s particular need for protection due to mental illness. At the same time, careless or even inflationary diagnoses must be counteracted, which can be ensured by guideline-based standards and meeting the minimum professional requirements for diagnosticians (e.g., psychotherapists and specialists in psychiatric/psychotherapeutic fields). This is the only way to ensure the best possible healthcare for the people concerned.

Human rights are not selectively applicable to individual groups. “Human rights are universal—or they are not. […] To be a bearer of human rights requires only one condition: belonging to the species” [30]. Because of this normative universalism of human rights, the exclusion of individuals from necessary medical care can only ever be unfounded.

## Data Availability

Not applicable.

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
