# Peer review of "Does the Human Right to Healthcare Apply Universally? A Contribution from a Trauma Therapeutic Perspective"

_ijerph, 2023, doi:10.3390/ijerph20156492_

Round 1
Reviewer 1 Report
The paper looks like a bit naive. Sure health care should be universal but costs are not. Should welfare states pay for the beneficiaries from their own citizens or we. need to build an universal welfare system? Authors are not providing any linkage to economy of health or to politics of health and they should. otherwise the full article is just a statement based in science.
Author Response
Dear reviewer,
We would like to express our gratitude for taking the time to review our manuscript. Thank you for your valuable feedback and pointing out the possible consequential costs that may arise from our human rights assessment. We have added a corresponding paragraph without conducting economic calculations, as this is beyond our expertise. We appreciate your overall positive evaluation of the essay.
Once again, thank you for your constructive criticism and support.
Sincerely, Bernd Hanewald
Reviewer 2 Report
have pointed out tin the pdf

Author Response
Dear reviewer,
We would like to express our gratitude for taking the time to review our manuscript. The feedback is invaluable in strengthening our manuscript and making the message clearer for the readers. We have taken your many excellent comments and feedback into account and made the necessary amendments to the manuscript. However, regarding the sample size, methods, etc., we were unable to implement your suggestions as this is not an empirical study but rather theoretical considerations illustrated with a case study, and which should therefore appear in the Opinion section of the Journal. Nevertheless, we want to emphasize that we appreciate your comments for improving the methodology.
Furthermore, we would like to express our gratitude for the literature references you provided. We have gladly incorporated them into the essay, and they have helped to enhance the quality of the text.
Once again, thank you for your constructive criticism and support.
Sincerely, Bernd Hanewald

Reviewer 3 Report
This article critiques the universality of the right to health care using a case study approach with a specific focus on the mental health of Ms B. In my opinion, this paper neatly presents migration as a social determinant of mental health, illustrating the process of migration and its relationship with mental health of migrants as well as the destructive and chaotic policies of the host countries (or destination countries) migrants find themselves in. This paper is a critical addition to the scholarly body of work on this topic.
The background neatly outlines the framework for the right to health care; with the policy and regulations of Germany through its Asylum Seekers Benefits Act. I do think there are instances where the authors can be more specific in their presentation of the framework and the AsylbLG, line 60 page 2, "...the presentation of acute symptoms ..." is this at Emergency OPDs or other emergency medical services. Also, later on in line 70 where the authors describe a discretionary leeway - is that in respect of asylum seekers presenting with acute illnesses and pain conditions? I'm also suggesting that the authors consider beginning the introduction with the prevalence of mental illness among refugees; and then move into the framework and the severe limitations of the AsylbLG. The loop is then closed when the authors present the aim of the paper = lines 158-160), which then leads to the case study.
The case study is presented in great detail (and with care) to adequately bring across Ms B's history. This presentation then culminates in the universality of human rights and its relevance for practice.
Author Response
Dear Reviewer,
We would like to express our gratitude for taking the time to review our manuscript. The feedback is invaluable in strengthening our manuscript. We have made the suggested clarifications in the text and have now placed the prevalence rates ahead of the contextual information. This change enhances the clarity and comprehensibility of the text. We greatly appreciate your constructive criticism and thank you for your support in improving our work.
Best regards,
Bernd Hanewald

Reviewer 4 Report
A good read. An important issue of migration health is discussed in-depth.
Few comments --- for more readability
-line 33 ... the number 1 is for reference... make it superscript
- line 46 and other, "[...]" might need clarification ...
- in the case vignette ... how long was the migration journey? time vs PTSD effect?
- migration transition phase effect? how long? Any events, such as imprisonment, trafficking, or any form of assault?
- where were the children born? timewise, before or after migration? after how long since arrival did she report to the Uni Hospital?
- relatives and friends and their role? could have some contribution, if she has migrated from an extended family system culture...
- line 187... introduce SUVs abbreviation
re-reading would suffice.
Author Response
Dear Reviewer,
We would like to express our gratitude for taking the time to review our manuscript. The feedback is invaluable in strengthening our manuscript and making the message clearer for the readers. We have taken your excellent comments and feedback into account and made the necessary amendments to the manuscript. We were happy to incorporate them into the paper, and they helped to improve the quality of the text.
Thank you again for your constructive criticism and support.
Yours sincerely, Bernd Hanewald

Round 2
Reviewer 2 Report
accepted
Author Response
Dear Reviewer,
I would like to sincerely thank you for your thorough review of our manuscript and for the valuable feedback you have provided. Your comments and suggestions were extremely helpful and undoubtedly contributed to improving the quality of our work. Also, thank you for recommending our manuscript for publication.
Best regards,
Bernd Hanewald